# Towards Reusable Ontology Alignment for Manufacturing Maintenance

Marco Kainzner, Christoph Klösch, Dominik Filipiak[0000−0002−4927−9992], Tek Raj Chhetri[0000−0002−3905−7878], Anna Fensel[0000−0002−1391−7104]

Semantic Technology Institute, Department of Computer Science, University of Innsbruck
{marco.kainzner,christoph.kloesch}@student.uibk.ac.at,
{dominik.filipiak,tekraj.chhetri,anna.fensel}@sti2.at

**Abstract.** With advancements in technology and big data availability, industries are struggling with data interoperability and knowledge representation. Ontologies have a great potential to solve such problems. However, lack of standardisation prevents the widespread adoption of ontologies in different manufacturing domains. Therefore, we investigate the possibility of preparing ontology alignment for manufacturing. This paper provides an overview of the available ontologies in this domain. We also provide an openly available alignment between two popular ontologies: IMAMO (maintenance ontology) and CDM-Core (process ontology).

**Keywords:** Manufacturing maintenance · Ontology alignment

## 1 Introduction

Ontologies provide the ability to model and represent knowledge in a reusable manner. They enable interoperability [8] and therefore have a high potential to improve processes and save costs in a variety of industries [6,3,10]. However, most of the ontologies are developed independently, which makes them incompatible, non-shareable, and severely limits their potential applications [5]. These issues could have been mitigated with more holistic or standardised development process. However, an alignment of independent ontologies can still overcome the aforementioned problems up to a certain extent. This paper presents our work towards such an alignment focused on maintenance. In order to combine ontologies that do not use common base in manufacturing, one has to survey existing ontologies in this domain. Therefore, we prepared the following research questions: *What ontologies for the manufacturing domain are available* (RQ1)? *To what extent can selected ontologies in the manufacturing maintenance domain be combined* (RQ2)?

The remainder of this paper is organised as follows. Section 2 provides an overview of the existing ontologies and answers RQ1. Section 3 explains the details of the ontology alignment process, which is related to RQ2. In Section 4, we discuss possible use cases for our work. The paper is concluded in Section 5.

## 2   Ontologies for manufacturing

In order to provide an ontology alignment, one has to identify ontologies suitable for this process. This section gives an overview of ontologies relevant to the manufacturing domain.

There are a number of ontologies dedicated to general manufacturing. For example, *Process Specification Language* (PSL) [4] has been built to "facilitate correct and complete exchange of process information among manufacturing system" [4]. PSL has been formalized in OWL. It is a NIST standard that is openly available online. The *CDM-Core* ontology [11] was developed as a common base ontology for the manufacturing domain. It is used for process models, services and sensor data. Its authors describe CDM-Core as "the first publicly available applied manufacturing ontology" [11]. *Additive Manufacturing Ontology* (AMU) was developed to address the lack of ontologies that are suited for modern manufacturing processes such as additive manufacturing [13]. It is developed as part of the Industrial Ontologies Foundry (IOF) initiative. Other manufacturing-related ontologies are *MASON* [9], *Machine Tool Model* (MTM) [7], *Machine of a process ontology* (MOP) [14], *Manufacturing Service Description Language* (MSDL) [1], and *Part-Focused Manufacturing Process Ontology* (PMPO) [12]. Most of these ontologies are upper ontologies or are designed for general manufacturing.

There are, however, some more specialised ontologies – for instance, developed specifically for manufacturing maintenance. IMAMO (*Industrial Maintenance Management Ontology*) [6] is designed to cover all aspects related to manufacturing maintenance. This ontology includes a variety of concepts related to the structure of equipment to be maintained – spare parts, failure detection, events, material resources, maintenance actors, technical documents, equipment states, and equipment life cycle. Another manufacturing maintenance ontology, *ROMAIN* [5] is similar in scope to IMAMO. It is built on top of the common Basic Formal Ontology (BFO) [2].

## 3   Alignment

In this section, we explain our choice of ontologies and detail the process of aligning them. We have chosen IMAMO (maintenance ontology) and CDM-Core (process ontology). These two ontologies were the best candidates for alignment for two main reasons. First, both ontologies seem to cover the subject of manufacturing well. Most of the other ontologies we examined are either upper ontologies or focus on more narrow disciplines within manufacturing. The scopes of IMAMO and CDM-Core are not equivalent, though – they are rather complementing each other. IMAMO concentrates on the maintenance process and defines some basic concepts for sensor data, whereas CDM-Core allows user to annotate process models, services and sensor data. The second reason for such choice is the fact that both ontologies are openly available online in OWL. Many other ontologies that we identified were not available.

The IMAMO ontology has 434 classes and contains 36 individuals, whereas the CDM-Core contains 240 classes and represents 18 individuals. In order to perform the mapping, we analysed each concept of CDM-Core and searched for corresponding semantic concepts in IMAMO. A manual alignment was performed, in which only the super classes of CDM-Core were considered. Since the overall number of classes was relatively small, no specialised tool for the alignment was needed. If we found an equivalent class, we created a `equivalentClass` element in our alignment. We were able to align 21 classes out of 27 super classes of CDM-Core. Some classes could not be matched. IMAMO defines a more granular monitoring systems (in computational resource): CMMS, Data Acquisition System, Diagnostics System, DMS etc. These classes are not present in CDM-Core, because it focuses more on the process modelling part. Additionally, IMAMO defines "external resource", which is used for defining subcontractors. In CDM-Core ontology, there is no such concept. The alignment is available publicly[1].

## 4   Possible Use Cases

There are several possible use cases for the presented alignment. One of these considers sensor data. The IMAMO ontology and the CDM-Core ontology have both sensor data defined. IMAMO defines IMAMO#Sensor and CDM-Core#Sensor. The IMAMO sensor is defined as a device that detects and responds to some input from the environment. In a manufacturing case, this would be the physical environment where the sensor is attached. With this definition, one can now use the CDM sensors (e.g. Electric power sensor, Pressure sensor etc.) in the IMAMO ontology without redefining it.

Another possible use case considers event-oriented systems. A key concept in maintenance is a triggering system that starts particular actions. IMAMO contains different classes facilitating this task: Alarm, Event Observed by User, Improvement Request, or Notification (RUL, Warning). CDM-Core defines the Component Fault class, which defines multiple faults (Gas Leakage, Cooler Efficiency Degradation). In IMAMO, one would model these cases with a Triggering Event – Alarm. With the provided alignment, one can react on CDM-Faults and trigger a Maintenance with IMAMO.

## 5   Conclusion

In this paper, we have given an overview of the diverse landscape of ontologies in the manufacturing domain. We analysed them in terms of a possibility of alignment. In conclusion, we decided to align the two publicly available ontologies CDM-Core and IMAMO fist. We were able to match 21 out of 27 root classes of CDM-Core with IMAMO. These alignments can act as a starting point for other researchers to publish more ontology alignments, and thereby make knowledge

---

[1] Link for the review process: `https://pastebin.com/PA2Pb9Wu`, password: `5mqAfuQcWa`. If accepted, it will be published publicly.

sharing in manufacturing easy and automated. Future work could build on our mapping and expand it by both adding more ontologies and more sub-concepts. Additionally, the SKOS data model could be used to enable more fine-grained alignment of concepts.

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
