# OpenReview forum: "Towards Reusable Ontology Alignment for Manufacturing Maintenance"
_eswc-conferences.org/ESWC/2021/Conference/Poster_and_Demo_Track — Submitted to ESWC2021 P&D_

### Official Review · AnonReviewer3 · 2021-04-13
**Ontology alignment of manufacturing ontologies, but where is the motivating application?**

**Rating:** 5
**Confidence:** 3

**Review:**

This paper makes a survey of the available ontologies for manufacturing maintenance and proposes an alignment for the two more popular ones, the IMAMO maintenance ontology and the CDM-Core process ontology. The alignment is made available, containing 20 equivalence links between classes. I wonder how deeply the alignment has been reviewed, since apparently conceptually far classes like Symptom and Trouble have been defined to be equivalent. However, what I really miss is a deeper motivation, not verbal but in the form of an actual application in the manufacturing domain, which uses either the alignment or some of the individual ontologies.

**Anonymity:**

Yes, I would like my review to remain anonymous.

---

### Official Review · AnonReviewer1 · 2021-04-14
**The introduced alignment is not evaluated**

**Rating:** 5
**Confidence:** 4

**Review:**

The paper motivates the need for alignment between ontologies for manufacturing, especially for applications in asset maintenance. Its main contribution is said to be an alignment between CDM-Core and IMAMO. However, the alignment is minimal (it includes equivalences only, no subsumption and no intermediary classes) and limited with respect to the size of the original ontologies (240 and 434 classes for 21 equivalences). Most importantly, the alignment is not evaluated: no competency question and no reasoning task have been defined.

The alignment focuses on alignment upper classes in CDM-Core (27 classes) but I don't understand the rationale: since IMAMO seems to be a specialization of CDM-Core (because it targets a certain domain of application), shouldn't the restriction rather be on the upper classes of IMAMO? CDM-Core being "above" IMAMO, overlap is rather between the lower part of CDM-Core and the upper part of IMAMO. Regardless, there are two few details on both ontologies in the paper to understand the value of the alignment.

Side nodes:
- the paper opens with a statement suggesting that ontologies are not compatible because they are independently developed. It is not necessarily true. In fact, there has been a latent hope in the Semantic Web community that is _not_ true in the general case. If ontologies were to be developed in a holistic way, there would be no need for a 'Web' ontology language, as a language to interlink conceptual models from various origins.
- LOV4IoT [1] includes several other ontologies for manufacturing. Have they been reviewed?
- a comparative table in section 2 would help. That table could e.g. show which ontology is publicly available and which one isn't.
- the sentence "both ontologies seem to cover the subject of manufacturing well" is an unusual statement for a research paper.

[1] http://lov4iot.appspot.com/?p=lov4iot-manufacturing

**Anonymity:**

Yes, I would like my review to remain anonymous.

---

### Official Review · AnonReviewer4 · 2021-04-15
**A manual alignment of manufacturing ontologies - lacking detail in description**

**Rating:** 6
**Confidence:** 5

**Review:**

The poster presents the manual alignment between two popular ontologies: IMAMO (maintenance ontology) and CDM-Core (process ontology) as well as a brief overview of existing ontologies in the domain.

The paper is lacking a description of how the alignment was performed. Specifically, how many people did the alignment? Where the people involved in the alignment domain experts? Was there an independent validation? Was the information available in the ontologies sufficient to perform the alignment, or was domain/external knowledge necessary?

All mappings in the alignment file are equivalences. Is this the full extent of the alignment? Are there no cases of subsumption or complex mappings? From the description of the ontologies, it appears there may be ample opportunity for subsumption mappings.

To increase the relevance for the SW community consider:
- explaining the use cases of the ontologies, and how their alignment supports extending them.
- showing specific examples of mappings that may be difficult to find in an automated fashion.
- Include subsumption mappings if they exist
- provide the alignment in the Alignment API format
https://www.w3.org/2001/sw/wiki/Alignment_API


**Anonymity:**

Yes, I would like my review to remain anonymous.

---

### Official Review · Program_Chairs · 2021-04-19
**Metareview: Reject (unclear motivation, no evaluation and shallow alignment)**

**Rating:** 5
**Confidence:** 5

**Review:**

This was a borderline paper, with reviewers leaning slightly towards reject, and slightly towards accept. The reviewers quote a lack of details regarding the overall alignment process, a lack of evaluation, and questions regarding the motivation. They also find the alignment to be quite shallow, featuring 20 equivalence links without subsumption or other forms of relations. Overall, with unclear motivation, no evaluation, and a shallow alignment whose process is not described, it would appear the contribution is currently not sufficient for acceptance in this track.

**Anonymity:**

Yes, I would like my review to remain anonymous.

---

### Decision · Program_Chairs · 2021-04-19

Reject